# Safety and Immunogenicity of the BBIBP-CorV Vaccine in Adolescents Aged 12 to 17 Years in the Thai Population: An Immunobridging Study

**DOI:** 10.3390/vaccines10050807

**Published:** 2022-05-19

**Authors:** Kriangkrai Tawinprai, Taweegrit Siripongboonsitti, Thachanun Porntharukchareon, Preeda Vanichsetakul, Saraiorn Thonginnetra, Krongkwan Niemsorn, Pathariya Promsena, Manunya Tandhansakul, Naruporn Kasemlawan, Natthanan Ruangkijpaisal, Narin Banomyong, Nanthida Phattraprayoon, Teerapat Ungtrakul, Kasiruck Wittayasak, Nawarat Thonwirak, Kamonwan Soonklang, Gaidganok Sornsamdang, Nithi Mahanonda

**Affiliations:** 1Department of Medicine, Chulabhorn Hospital, Chulabhorn Royal Academy, Bangkok 10210, Thailand; taweegrit.sir@cra.ac.th (T.S.); thachanun.por@cra.ac.th (T.P.); nithi.mah@cra.ac.th (N.M.); 2Department of Pediatrics, Chulabhorn Hospital, Chulabhorn Royal Academy, Bangkok 10210, Thailand; preeda.van@cra.ac.th (P.V.); saraiorn.tho@cra.ac.th (S.T.); krongkwan.nie@cra.ac.th (K.N.); pathariya.pro@cra.ac.th (P.P.); manunya.tan@cra.ac.th (M.T.); naruporn.kas@cra.ac.th (N.K.); natthanan.rua@cra.ac.th (N.R.); narin.ban@cra.ac.th (N.B.); 3Princess Srisavangavadhana College of Medicine, Chulabhorn Royal Academy, Bangkok 10210, Thailand; nanthida.pha@cra.ac.th (N.P.); teerapat.ung@cra.ac.th (T.U.); 4Center of Learning and Research in Celebration of HRH Princess Chulabhorn’s 60th Birthday Anniversary, Chulabhorn Royal Academy, Bangkok 10210, Thailand; kasiruck.wit@cra.ac.th (K.W.); nawarat.tho@cra.ac.th (N.T.); kamonwan.soo@cra.ac.th (K.S.); 5Central Laboratory Center, Chulabhorn Hospital, Chulabhorn Royal Academy, Bangkok 10210, Thailand; gaidganok.sor@cra.ac.th

**Keywords:** COVID-19, SARS-CoV-2, adolescent, BBIBP-CorV, safety, immunogenicity

## Abstract

Adolescents can develop a severe form of Coronavirus disease 2019 (COVID-19), especially with underlying comorbidities. No study has examined the efficacy or effectiveness of inactivated COVID-19 vaccines in adolescents. This single-center, prospective cohort study was performed to evaluate the safety and effectiveness of an inactivated COVID-19 vaccine in adolescents using the immunobridging approach at Chulabhorn Hospital. The key eligibility criterion was a healthy clinical condition or stable pre-existing comorbidity. The anti-receptor-binding domain (anti-RBD) antibody concentration at 4 weeks after dose 2 of the vaccine was compared between participants aged 12 to 17 years and those aged 18 to 30 years. Safety profiles included adverse events within 7 days after each dose of the vaccine and any adverse events through 1 month after dose 2 of the vaccine. In the adolescent and adult cohorts, the geometric mean concentration of anti-RBD antibody was 102.9 binding antibody unit (BAU)/mL (95% CI, 91.0–116.4) and 36.9 BAU/mL (95% CI, 30.9–44.0), respectively. The geometric mean ratio of the adolescent cohort was 2.79 (95% CI, 2.25–3.46, *p* < 0.0001) compared with the adult cohort, meeting the non-inferiority criterion. The reactogenicity was slightly lower in the adolescent than in the adult cohort. No serious adverse events occurred. The inactivated COVID-19 vaccine appears safe and effective in adolescents.

## 1. Introduction

The COVID-19 pandemic has affected all populations worldwide, including adolescents. In general, most children and adolescents develop a mild form of COVID-19 [1]. However, some adolescents can develop a severe form of COVID-19, especially adolescents with underlying comorbidities [2]. In addition, there have been reports of children and adolescents developing a condition called multisystem inflammatory syndrome in children (MIS-C), which is a complication related to severe acute respiratory syndrome coronavirus 2 (SARS-CoV-2) infection [3]. MIS-C affects multiple organ systems, especially exhibiting cardiovascular and mucocutaneous involvement and extreme inflammation [4]. Prior studies in adults have shown that COVID-19 vaccines can prevent symptomatic SARS-CoV-2 infection, hospitalization, and death [5,6,7,8]. However, studies of the safety and effectiveness of COVID-19 vaccines in children and adolescents are limited. Two studies have focused on the BNT 162b2 vaccine in children aged 5 to 11 years and 12 to 15 years. These studies showed a favorable safety profile and immunogenicity compared with the adult cohort [9,10]. The BNT162b2 is an mRNA vaccine platform, developed by BioNTech and Pfizer [11]. The U.S. Food and Drug Administration authorized the emergency use of the BNT162b2 vaccine for the prevention of COVID-19 in children and adolescents ages 5 years and older [12].

The BBIBP-CorV vaccine is an inactivated COVID-19 vaccine developed by the Beijing Bio-Institute of Biological Products (BBIBP), a China National Biotec Group (CNBG) subsidiary. The China National Pharmaceutical Group Corporation (Sinopharm) is CNBG’s parent company [13]. This vaccine was prepared from whole SARS-CoV-2 (HB02 strain), inactivated with β-propiolactone, containing 4 μg of SARS-CoV-2 virion and 0.45 mg of alum [14,15]. A phase 3 clinical trial demonstrated efficacy and safety in participants aged ≥18 years. The efficacy for preventing symptomatic COVID-19 was high, reaching 78.1% [8]. In addition, a community-based, observational study included 214,940 PCR-positive cases of COVID-1 in Abu Dhabi Emirate, United Arab Emirates (UAE), reported that vaccine effectiveness in fully vaccinated individuals with BBIBP-CorV was 80% and 97% to prevent COVID-19-related hospital admissions and death, respectively [16]. Moreover, BBIBP-CorV was one of the COVID-19 vaccines in the World Health Organization Emergency Use Listing [17] and authorized by 45 countries for use in adults ≥ 18 years [18]. The safety profile after vaccination in children and adolescents aged 3 to 17 years in phase ½ trials was favorable. In participants aged 13 to 17 years, the local and systemic solicited adverse reaction was reported as only 11.5% (29/252) and 22.6% (57/252), respectively. Most adverse reactions were mild to moderate in severity, and no serious adverse event was reported. The immune response was lowest in the participants who received 2 μg of the BBIBP-CorV vaccine. There was no difference in immune response between the participants who received 4 μg or 8 μg of the vaccine [19]. However, studies on vaccination effectiveness in adolescents are lacking.

An immunobridging study for COVID-19 vaccine effectiveness is acceptable in the children and adolescent group. The non-inferiority of immune response compared with the adult population with known vaccine efficacy inferred vaccine effectiveness [9,10]. This study was performed to evaluate the safety profile and effectiveness of the BBIBP-CorV vaccine with the immunobridging approach in Thai adolescents aged 12 to 17 years compared with adults aged 18 to 30 years.

## 2. Materials and Methods

### 2.1. Trial Design and Participants

We conducted a single-center, open-label, prospective cohort study to compare the immunogenicity and safety of the BBIBP-CorV vaccine between participants aged 12 to 17 years (adolescent cohort) and participants aged 18 to 30 years (adult cohort) at Chulabhorn Hospital, Chulabhorn Royal Academy, Bangkok, Thailand. The key eligibility criterion for the adolescent cohort was a healthy clinical condition or stable pre-existing comorbidity. The exclusion criteria were pregnancy, lactation, fever or respiratory tract infection, or receipt of any vaccine within 14 days of enrolment. Participants with known previous SARS-CoV-2 infection or prior COVID-19 vaccination were also excluded. The data of the adult cohort were obtained from a prior safety and immunogenicity study of the BBIBP-CorV vaccine in adult participants in our institute. Informed consent was obtained from all participants (and their parents, if applicable) involved in the study. The study was conducted according to the guidelines of the Declaration of Helsinki. The study protocol, informed consent, and case record form were reviewed and approved by the Ethics Committee for Human Research, Chulabhorn Research Institute (reference number: 123/2564). The study was registered with thaiclinicaltrials.org (TCTR20210920005).

### 2.2. Interventions

All participants, both adolescents and adults, received two doses of 4 μg (0.5 mL) of BBIBP-CorV vaccine intramuscularly in the deltoid muscle, at a 21-day interval [19] at the Chulabhorn Royal Academy COVID-19 vaccine clinic. The well-trained registered nurses administered the study vaccine to all participants. After each vaccination, all participants were observed for the development of adverse events (AEs) at the clinic for 30 min.

### 2.3. Safety

The local and systemic reactogenicities were evaluated using a questionnaire sent by mobile text message on days 1 and 7 post-vaccination for each dose. In addition, unsolicited and serious AEs were followed up for 1 month using the mobile text message questionnaire on day-30 post-vaccination and a mobile hotline for participants to report their AEs. Pain, tenderness, redness, and swelling were defined as local reactogenicity. Fever, nausea or vomiting, diarrhea, headache, fatigue, and myalgia were defined as systemic reactogenicity. Local AEs were graded as mild if the lesion was less than 5 cm, moderate if the lesion was 5 to 10 cm, severe if the lesion was more than 10 cm, and life-threatening if an emergency department visit or hospitalization was required. In addition, a systemic AE was graded as mild if the AE did not interfere with daily activity, moderate if the AE interfered with some daily activity, severe if the AE limited daily activity, and life-threatening if an emergency department visit or hospitalization was required.

### 2.4. Immunogenicity

Immunogenicity was assessed by measuring the concentration of anti-receptor-binding domain (RBD) of the spike protein of SARS-CoV-2 before the first vaccination and 1 month after the second vaccination using an Elecsys Anti-SARS-CoV-2 S (Elecsys-S) kit (Roche Diagnostics, Mannheim, Germany), an automated electrochemiluminescence immunoassay. The test was performed in accordance with the manufacturer’s instructions. The concentration of anti-RBD antibodies is presented in this report as the binding antibody units per milliliter (BAU/mL) following the World Health Organization international standard for anti-SARS-CoV-2 immunoglobulin titers [20]. The Elecsys-S units were converted to BAU as follows: Elecsys-S U = 0.972 × BAU [21].

### 2.5. Outcomes

The primary outcome was the geometric mean concentration (GMC) of anti-RBD antibody 4 weeks after the second vaccination between the adolescent and adult cohorts. In addition, a non-inferiority comparison of the anti-RBD antibody GMC between the adolescent and adult cohorts was performed, and the result is shown using the geometric mean ratio (GMR).

The two secondary outcomes were reactogenicity within 7 days after the BBIBP-CorV vaccination and serious AEs within 1 month after the second vaccination.

### 2.6. Sample Size

A sample of 250 participants in each cohort was estimated to provide 80% power, 0.05 type I error, and 25% dropout for declaring non-inferiority [10]. Non-inferiority was defined as the lower limit of the two-sided 95% confidence interval of a GMR > 0.67.

### 2.7. Statistical Methods

The mean of anti-RBD antibody concentration was demonstrated as geometric mean concentration (GMC).

A comparison of the anti-RBD antibody concentration between the adolescent and adult cohorts was performed using an independent-samples *t*-test if the data were normally distributed. On the other hand, the Mann–Whitney U test was used to compare data that were not normally distributed. The distribution of the data was analyzed using the Shapiro–Wilk test. The result of the comparison was demonstrated as GMR with a 95% confidence interval (CI). If the non-inferiority criterion was met, then superiority was assessed. Summary statistics are presented as median and interquartile range or GMC and 95% confidence interval (CI). Statistical analyses were performed using GraphPad Prism version 9 (GraphPad Software, San Diego, CA, USA) and SPSS version 26 (IBM Corp., Armonk, NY, USA). A *p*-value of <0.05 was considered statistically significant.

## 3. Results

### 3.1. Participants

From 14 September 2021 to 5 October 2021, 248 adolescent participants were assessed for eligibility. Fourteen participants were excluded (five because of a seropositive status at baseline and nine because of the inability to evaluate the baseline serostatus). For the adult cohort, 252 participants aged 18 to 30 years were assessed for eligibility. Three were excluded because of a seropositive status at baseline. Finally, 234 participants in the adolescent cohort and 249 in the adult cohort received the study vaccine and underwent the safety profile analysis. All participants in the study received both doses of the study vaccine. However, 4 weeks after the second vaccination, only 393 (81.4%) participants were followed up for the primary outcome analysis (190 in the adolescent cohort and 203 in the adult cohort). The participant flow chart is shown in Figure 1.

The baseline demographic data are summarized in Table 1. In the adolescent cohort, almost half of the participants (46.2%) were female, and the median age was 14 (13–16) years. In the adult cohort, 47.4% were female, and the median age was 25 (2–28) years. No participants had underlying comorbidity.

### 3.2. Safety

#### 3.2.1. Reactogenicity

The reactogenicity after the BBIBP-CorV vaccination in the adolescent and adult cohorts is summarized in Figure 2 and Figure 3, respectively. The reactogenicity was slightly lower in the adolescent than in the adult cohort. In both cohorts, the reactogenicity was somewhat higher after the first than second dose. On day 1 after the first dose, the reactogenicity was 18.1% in adolescents and 23.7% in adults. Most reactions were mild to moderate in severity and resolved within 7 days. One participant in the adolescent cohort reported severe nausea and vomiting that spontaneously resolved without a hospital visit.

Seven participants in the adult cohort reported nine severe reactogenicities (injection site reaction in one participant, headache in one, fatigue in one, myalgia in one, and drowsiness in five). However, all symptoms resolved within 1 week. On day 1 after the second dose, the reactogenicity was 12.4% in the adolescent cohort and 14.1% in the adult cohort. All participants’ reactions were mild to moderate in severity except one reaction in the adult cohort; this participant reported severe myalgia that spontaneously resolved. No serious AEs were reported during the follow-up period.

Local reactions were more common than systemic reactions in the adolescent cohort. After the first vaccination, the most common local reaction was pain at the injection site (13.2%). The three most common systemic reactogenicities were fatigue (8.7%), myalgia (7.7%), and headache (7.7%). In contrast to the adolescent cohort, local reactions were less common than systemic reactions in the adult cohort. After the first vaccination, the three most common systemic reactogenicities were drowsiness (15.6%), myalgia (12.8%), and fatigue (11.6%). Local reactions were reported in only 7.2% of adult participants.

#### 3.2.2. Adverse Events

AEs were reported in approximately 3.4% of the adolescent cohort and 2.0% of the adult cohort. No vaccine-related AEs were reported in the adolescent cohort; however, vaccine-related AEs were reported in 0.8% of the adult cohort. No severe, serious, or life-threatening AE; death; or AE leading to discontinuation was reported in either cohort (Table 2).

### 3.3. Immunogenicity

Four weeks after the second vaccination, the GMC of anti-RBD antibody in the adolescent cohort was 102.9 BAU/mL (95% CI, 91.0–116.4), and that in the adult cohort was 36.9 BAU/mL (95% CI, 30.9–44.0) (Figure 4). The GMR in the adolescent cohort was 2.79 (95% CI, 2.25–3.46; *p* < 0.0001) compared with the adult cohort (Table 3). The immune response in the adolescent cohort met the non-inferiority criterion compared with the adult cohort (lower limit of two-sided 95% CI > 0.67). Moreover, the lower limit of the two-sided 95% CI was >1, indicating superiority of the immune response in the adolescent cohort compared with the adult cohort. The subgroup analysis also showed that the GMR of adolescents aged 12 to 14 years was 3.37 (95% CI, 2.59–4.38; *p* < 0.0001), and that of adolescents aged 15 to 17 years was 2.27 (95% CI, 1.74–2.98; *p* < 0.0001) (Table 4). The per-protocol analysis including participants with unknown serostatus at baseline also showed a similar result (Table 3 and Table 4).

## 4. Discussion

Two doses of the BBIBP-CorV vaccine at a 21-day interval can elicit a more robust immune response in adolescents aged 12 to 17 years than in adults aged 18 to 30 years at 1 month after the second dose with similar safety profiles. A robust immune response was observed in both younger adolescents (aged 12–14 years) and older adolescents (aged 15–17 years). The phase 3 trial of the BBIBP-CorV vaccine showed high vaccine efficacy of 78.1% in participants aged ≥18 years [8]. The results of our study suggest the high effectiveness of the BBIBP-CorV vaccine in adolescents aged 12 to 17 years through the immunobridging approach. The safety profile after vaccination with the BBIBP-CorV vaccine was slightly lower in adolescents than in adults with respect to both reactogenicity and unsolicited AEs.

Our study results are similar to those of other studies focusing on the immunogenicity and safety of other vaccine platforms in adolescents. Frenck et al. [10] assessed the safety and immunogenicity of the BNT162b2 vaccine, an mRNA vaccine platform, in adolescents aged 12 to 15 years compared with participants aged 16 to 25 years. The AEs were comparable between the two age groups, and no vaccine-related serious AEs occurred in either cohort. The immunogenicity in participants aged 12 to 15 years was superior to that in participants aged 16 to 25 years, with a GMR of 1.76 (95% CI, 1.47–2.10). This finding was similar to our study. The GMR of anti-RBD antibody in the adolescents who received the BBIBP-CorV vaccine was 2.79 (95% CI, 2.25–3.46) compared with the adult cohort. Moreover, no participant who received BNT162b2 had developed COVID-19 by 7 days after the second vaccination. Conversely, 16 of 978 participants who received a placebo developed COVID-19. However, the reactogenicities in our adolescent cohort were lower compared with the BNT162b2 vaccine. Pain at the injection site was reported in 86% of the BNT162b2 vaccine group compared with 13.2% in our cohort. The systemic reactogenicity was also lower in the BBIBP-CorV vaccine. The three most common systemic reactogenicities in BNT162b2 vaccine were fatigue (60%), headache (55%), and chills (28%). On the other, the three most common systemic reactogenicities were fatigue (8.7%), myalgia (7.7%), and headache (7.7%) [10]. Similar results regarding safety and immunogenicity were demonstrated in participants aged 5 to 11 years after two doses of 10 μg of the BNT162b2 vaccine at a 21-day interval. In total, 1517 children were randomized to receive the BNT162b2 vaccine, and 751 received a placebo. The safety profile was similar to the other age group. No vaccine-related serious AEs were observed. The immunogenicity was non-inferior to that in adults aged 16 to 25 years. The GMR of neutralizing titer in children aged 5 to 11 years to that in participants aged 16 to 25 years was 1.04 (95% CI, 0.93–1.18), which met the non-inferiority criterion [9]. The other mRNA vaccine, mRNA-1273, also evaluated the safety and immunogenicity in adolescents aged 12 to 17 years. Like the adult participants, two doses of mRNA-1273 (100 μg in each) with a 28-day interval were provided intramuscularly. Both local and systemic reactogenicities were high compared with the BBIBP-CorV vaccine. The pain after injection was reported in 93.1% after the first dose and 92.4% after the second dose. The systemic reactogenicity was reported in 68.5% after the first dose and 86.1% after the second dose. The GMR of pseudovirus neutralizing antibody titers in adolescents compared with adults was 1.08 (95% CI, 0.94–1.92), which met the noninferiority criterion. In terms of immunogenicity, this study’s result differed a bit compared with our findings. Adolescents who received the BBIBP-CorV vaccine had an immune response more robust than adults (GMR: 2.79, 95% CI, 2.25–3.46) [22]. Due to the difference in immune response measurement, we could not compare the immune response directly with the of mRNA vaccines [10,22]. However, the immune response from the inactivated COVID-19 vaccine seems to be less than the mRNA vaccine in Khoury et al. finding [23].

Post-authorization vaccine effectiveness evaluations supported the immunobridging result of the BNT162b2 vaccine. The effectiveness of BNT162b2 against hospitalization in participants aged 12 to 18 years was also evaluated using a test-negative design in 19 pediatric hospitals. The case group comprised hospitalized patients with COVID-19-like symptoms with positive SARS-CoV-2 reverse-transcription polymerase chain reaction (RT-PCR) results. The two control groups were (1) hospitalized patients with COVID-19-like symptoms with negative SARS-CoV-2 RT-PCR or antigen test results (test-negative) and (2) hospitalized patients without COVID-19-associated symptoms. In total, 6 of 179 participants in the case group were vaccinated with BNT162b2, and 93 of 285 participants in the control group received BNT162b2. The vaccine’s effectiveness against hospitalization was 93% (95% CI, 83–97) [24].

The safety profile in our study was not different from that in the phase 2 clinical trial of the BBIBP-CorV vaccine. The phase 1/2 clinical trial of the BBIBP-CorV vaccine was performed to evaluate the safety profile and immunogenicity in participants aged 3 to 17 years. The participants were stratified according to age (3–5 years, 6–12 years, and 13–17 years) and dose (2 μg, 4 μg, and 8 μg) and were randomized to receive the vaccine or placebo. The vaccines were scheduled in three doses (days 0, 28, and 56). There were 288 participants in the phase 1 trial and 720 in the phase 2 trial. The adverse reactions were evaluated within 30 days after the whole vaccination procedure. The immunogenicity was assessed 28 days after dose 1, dose 2, and dose 3 of the BBIBP-CorV vaccine using the neutralizing antibody titer. The adverse reactions were mild to moderate in severity in most participants. Pain at the injection site was the most common local reaction in participants aged 13 to 17 years (7.9%), and fever was the most common systemic reaction (10.3%). With respect to the immunogenicity outcome, the immune responses in the 4-μg and 8-μg groups were higher than that in the 2-μg group. By day 28 after dose 2 in the 3- to 5-year-old cohort, the GMT was 105.3 (95% CI, 95.4–116.2) in the 2-μg group, 180.2 (95% CI, 163.4–198.8) in the 4-μg group, and 170.8 (95% CI, 202.2–249.1) in the 8-μg group. The GMT ranged from 84.1 to 168.6 in the 6- to 12-year-old cohort and from 88.0 to 155.7 in the 13- to 17-year-old cohort [11]. However, Xia et al. mention that the immunogenicity in adolescents was similar to the adult cohort [19]. The different findings of our study may occur due to the difference in the humoral immune response measurement. The study of Xia et al. used the neutralizing antibody test which was different from our study using the binding antibody test.

Our study also found that the younger participants had a more robust immune response than older or adult participants. This finding may be due to the weight dependence of the vaccine. This finding can also be found with the BNT162b2 vaccine, the participants aged 5–11 received the lower amount of vaccine but had a similar immune response [9].

To our knowledge, this is the first study to evaluate the effectiveness of the BBIBP-CorV vaccine using the immunobridging approach. The results of our study support the use of the BBIBP-CorV vaccine in adolescents aged 12 to 17 years. Although most adolescents have a mild form of COVID-19 [1], vaccination in this group also has benefits. First, vaccination can prevent severe COVID-19, especially in adolescents with underlying comorbidities [10,20,25]. Therefore, vaccination in this group should decrease morbidity and mortality. Second, evidence supports the notion that vaccination with an mRNA COVID-19 vaccine can prevent MIS-C, a severe complication associated with SARS-CoV-2 infection in children and adolescents. At the end of December 2021, 76.7% of adolescents in France had been vaccinated with at least one dose of the COVID-19 vaccine, mostly BNT162b2. In this period, 33 adolescents were hospitalized because of MIS-C. Among them, 7 had received a single dose of the vaccine and 26 were unvaccinated. No adolescent that was fully vaccinated with the COVID-19 vaccine developed MIS-C in this period. The hazard ratio for MIS-C in vaccinated participants was 0.09 (95% CI, 0.04–0.21) [3]. Third, the vaccination of adolescents can improve herd immunity for community prevention [24].

Moreover, unlike the other vaccine platform in which younger participants may develop severe AEs, no special AEs related to the BBIBP-CorV vaccine occurred in the present study. For example, after vaccination with the mRNA162b2, a small proportion of younger participants developed myocarditis, especially male participants [26,27,28]. Vaccine-induced thrombotic thrombocytopenia is associated with the adenoviral vector platform, especially in the young population [29].

This study had two main limitations. First, we conducted a prospective cohort study of adolescents and compared the results with those of our previous study of adult participants instead of performing a parallel clinical trial. Most adult participants were vaccinated when the study started. It was difficult to enroll adult participants without prior vaccination with the COVID-19 vaccine. Therefore, there were differences in the case record forms between the adolescent and adult cohorts. Only a single questionnaire for local reactions was used in the adult cohort. However, the other solicited AEs were the same. Second, we used text message questionnaires instead of a diary book. Some participants in the adolescent cohort did not respond to the safety questionnaires. This limitation may improve by active contact with participants who do not complete questionnaires.

## 5. Conclusions

Vaccination with the BBIBP-CorV vaccine in adolescent participants was safe and produced a more robust immune response than in adults aged 18 to 30 years.

## Figures and Tables

**Figure 1 vaccines-10-00807-f001:**
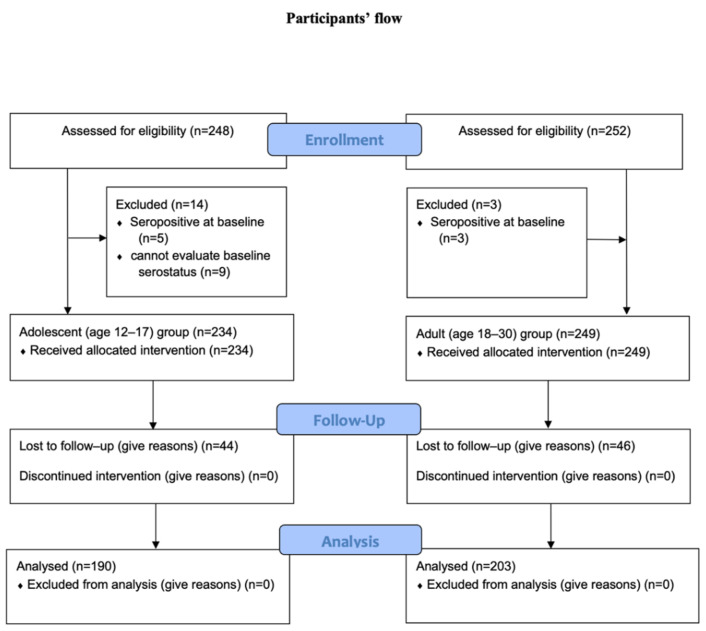
Flow chart of study participants. From 14 September 2021 to 5 October 2021, 248 adolescent participants were assessed for eligibility. Fourteen participants were excluded. Additionally, 252 participants aged 18 to 30 years were assessed for eligibility for the adult cohort. Three were excluded because of a seropositive status at baseline. Finally, 234 participants in the adolescent cohort and 249 in the adult cohort received the study vaccine and underwent the safety profile analysis. Four weeks after the second vaccination, 393 (81.4%) participants were followed up for the primary outcome analysis (190 in the adolescent group and 203 in the adult group).

**Figure 2 vaccines-10-00807-f002:**
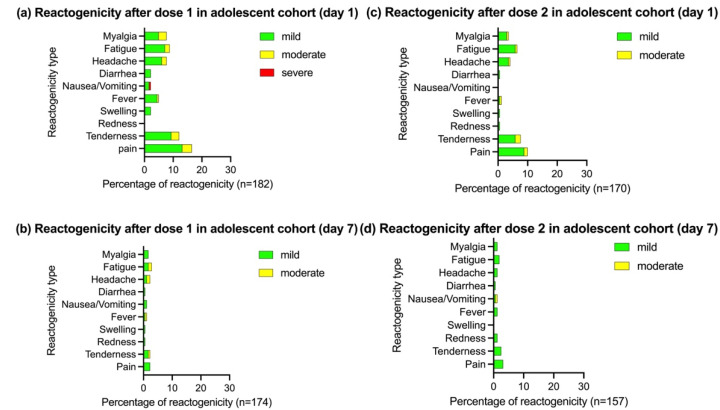
Reactogenicity on days 1 and 7 after the BBIBP-CorV vaccination in the adolescent cohort (age of 12–17 years). Pain, tenderness, redness, and swelling were defined as local reactogenicity. Fever, nausea or vomiting, diarrhea, headache, fatigue, and myalgia were defined as systemic reactogenicity. n: number of participants who responded to the questionnaire. (**a**) Reactogenicity on day 1 after dose 1. (**b**) Reactogenicity on day 7 after dose 1. (**c**) Reactogenicity on day 1 after dose 2. (**d**) Reactogenicity on day 7 after dose 2.

**Figure 3 vaccines-10-00807-f003:**
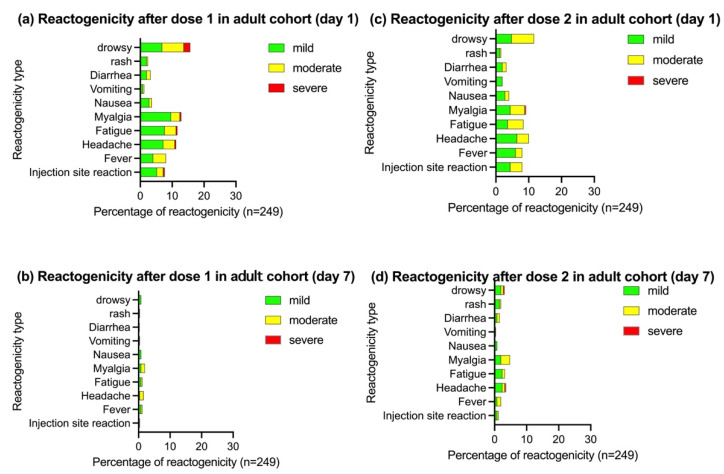
Reactogenicity on days 1 and 7 after the BBIBP-CorV vaccination in the adult cohort (age of 18–30 years). Pain, tenderness, redness, and swelling were defined as local reactogenicity. Fever, nausea or vomiting, diarrhea, headache, fatigue, and myalgia were defined as systemic reactogenicity. n: number of participants who responded to the questionnaire. (**a**) Reactogenicity on day 1 after dose 1. (**b**) Reactogenicity on day 7 after dose 1. (**c**) Reactogenicity on day 1 after dose 2. (**d**) Reactogenicity on day 7 after dose 2.

**Figure 4 vaccines-10-00807-f004:**
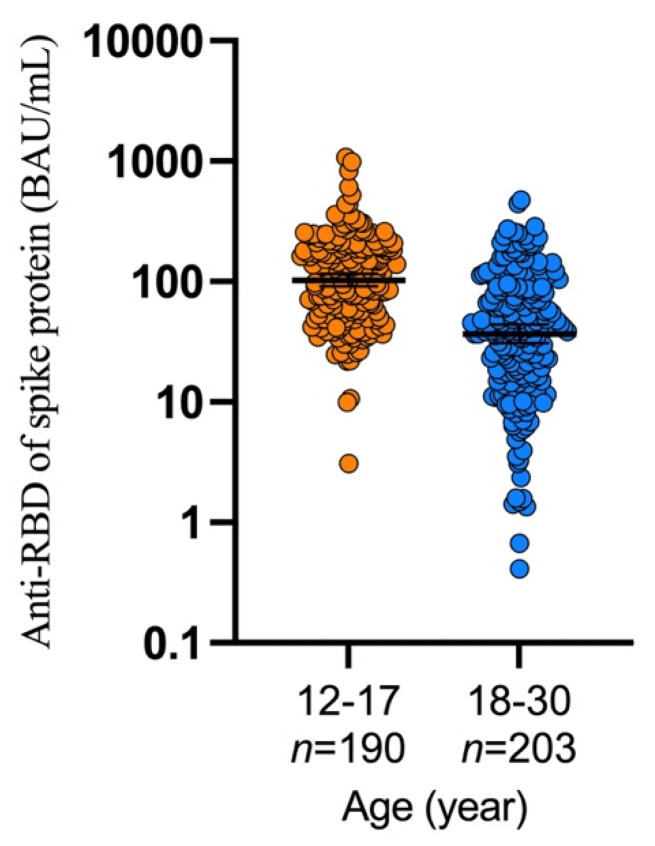
Concentration of anti-receptor binding domain of spike protein of SARS-CoV-2. BAU: binding antibody units, *n*: number of participants for immunogenicity analysis. *p* < 0.0001 (Mann–Whitney U test).

**Table 1 vaccines-10-00807-t001:** Demographic data of each study cohort.

	Adolescent Cohort 12–17 Years	Adult Cohort 18–30 Years
Participants in safety analysis	234	249
Female sex	108 (46.2)	118 (47.4)
Age, years	14 (13–16)	25 (22–28)
Participants in immunogenicity analysis	190	203
Female sex	90 (47.4)	97 (47.8)
Age, years	14 (13–16)	25 (23–28)
Underlying disease	0	0

Data are presented as number, *n* (%), or median (interquartile range).

**Table 2 vaccines-10-00807-t002:** Adverse events after dose 1 through 1 month after dose 2.

Adverse Event	Adolescents 12–17 Years Old (*n* = 234)	Adults 18–30 Years Old (*n* = 249)
	*n* (%)	*n* (%)
Any event	8 (3.4%)	5 (2.0%)
Related	0 (0%)	2 (0.8%)
Severe	0 (0%)	0 (0%)
Life-threatening	0 (0%)	0 (0%)
Any serious adverse event	0 (0%)	0 (0%)
Related		
Severe		
Life-threatening		
Any adverse event leading to discontinuation	0 (0%)	0 (0%)
Related		
Severe		
Life-threatening		
Death	0 (0%)	0 (0%)

**Table 3 vaccines-10-00807-t003:** GMC and GMR in each study group.

	12–17 Years	18–30 Years
Number of participants ^a^	190	203
GMC (95% CI), BAU/mL	102.9 (91.0–116.4)	36.9 (30.9–44.0)
GMR (95% CI)	2.79 (2.25–3.46) *	reference
Number of participants ^b^	198	203
GMC (95%CI) (BAU/mL)	102.49 (90.81–115.68)	36.9 (30.9–44.0)
GMR (95%CI)	2.78 (2.25–3.44) *	reference

^a^: included participants with seronegative status. ^b^: included 8 participants in adolescent cohort without known serostatus. GMC: geometric mean concentration, GMR: geometric mean ratio, CI: confidence interval, BAU: binding antibody units. * *p* < 0.0001 (Mann–Whitney U test).

**Table 4 vaccines-10-00807-t004:** GMC and GMR in each subgroup.

	12–14 Years	15–17 Years	18–30 Years
Number of participants ^a^	99	91	203
GMC (95% CI), BAU/mL	124.2 (105.8–145.7)	83.9 (69.8–100.7)	36.9 (30.9–44.0)
GMR (95% CI)	3.37 (2.59–4.38) *	2.27 (1.74–2.98) *	reference
Number of participants ^b^	103	95	203
GMC (95%CI) (BAU/mL)	125.3 (107.2–146.7)	82.4 (68.9–98.5)	36.9 (30.9–44.0)
GMR (95%CI)	3.4 (2.53–4.56) *	2.23 (1.65–3.02) *	reference

^a^: included participants with seronegative status. ^b^: included 4 participants in each adolescent subgroup without known serostatus. GMC: geometric mean concentration, GMR: geometric mean ratio, CI: confidence interval, BAU: binding antibody units. * *p* < 0.0001 (Mann–Whitney U test).

## Data Availability

The datasets generated and analyzed during the current study are available from the corresponding author upon reasonable request.

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
