# Peer review of "Safety and Immunogenicity of the BBIBP-CorV Vaccine in Adolescents Aged 12 to 17 Years in the Thai Population: An Immunobridging Study"

_vaccines, 2022, doi:10.3390/vaccines10050807_

Round 1

Reviewer 1 Report

This is a study on the effectiveness of an inactivated COVID-19 vaccine, called BBIBP-CorV, on adolescents. It is an important step in identifying the usefulness of another COVID-19 vaccine in protecting adolescents from the disease. The authors concluded that the BBIBP-CorV vaccine was safe to use in adolescents and induced a more robust immune response than adults aged 18-30 years.

There are some issues with grammar at points, where the phrases don’t make much sense, usually due to the wrong use of words, and it can, in some cases, be difficult to understand from the context on what the phrase is supposed to be saying. For example, in the first paragraph of the introduction, the second phrase mentions “most COVID-19 children or adolescents were a mild form of severity” where the word ‘were’ makes no sense here. Instead, it should have been something like ‘have’ or ‘suffer from’. The very next phrase has “especially with comorbidity underlying”, where the word ‘underlying’ should be before ‘comorbidity’, not after, so it should be ‘with underlying comorbidity’. There seems to be less of this problem in the discussion section but still an occasional error there. In addition, there are such grammar issues in the statistical methods.

This study deals a lot with statistical methods which need further explanations.  I found that most of the problems, however, do appear to be with the grammar and the resulting confusing phrases.

Reviewer 2 Report

-Page 2, Line 74-76:All participants were assigned to receive two doses of 4 mg (0.5 mL) of BBIBP-CorV vaccine intramuscularly, why you use 4ug NOT 2 or 8 ug please cite the study already evaluated this (Lancet Infect Dis) Ref 11 in your list!. AND e intramuscularly at 21 days intervals, Same Ref 11.

-Page 3, lines 100-107: please ref !! cite because you used the same parameters they already used.

-Although your study estimated that BBIBP-CorV was more immunogenic in 12-17 than the previous report (Ref 11), did you know why?, please add more discussion, also your results demonstrated that BBIBP-CorV was more immunogenic in 12-17 than adults aged 18-30 years at one month although the previous report (Ref.11) Say “which were similar to the BBIBP-CorV-elicited antibody level in adult participants” a discussion/clarification also need.

-As in table 4 “A robust immune response was observed in both younger (aged 12-14 years) and older (aged 15-17 years) if we add 18-30 years with immunity: GMC (95%CI) (BAU/mL) =124.2 (105.8-145.7) 83.9 (69.8-100.7) 36.9 (30.9-44.0), respectively, is this indicate to exponential decline overage? Please discuss the issue too, because the literature revealed, that when comparing BBIBP-CorV with mRNA vaccine, BBIBP-CorV efficacy is rapidly decline.

Reviewer 3 Report

Tawinprai et al. reported that Safety and immunogenicity of the BBIBP-CorV vaccine in adolescents aged 12-17 years in Thai population, a prospective cohort study.

  1. Authors should explain “BBIBP-CorV vaccine” when this first appears in abstract section. Do not use the abbreviation.
  2. Do not use the abbreviations, such as COVID-19, GMC, BAU, and GMR when they first appear in abstract section.
  3. In discussion section, authors should compare BBIBP-CorV vaccine with Pfizer and Moderna vaccines.
  4. Authors should explain more about BBIBP-CorV vaccine, because we do not know this vaccine well.

Reviewer 4 Report

in the introduction: why did the authors compare data among adolescents and adults? It is not clear. please specify

clearly report inclusion/exclusion criteria

more details about intervention should be added. as for instance place of injection, via of injection, who performed the injection and so on.

many details are missing, please refer to the consort guidelines to improve the reporting of your trial.

please, consider to report both intention to treat and per protocol analysis.

in the discussion section a comparison with other type of COVID-19 vaccines should be added both in terms of safety and immunogenicity.

In the funding statement please also add (if any) role in the design, performing, and reporting of the sponsor.

Round 2

Reviewer 1 Report

  • According to the authors' response, I won't able to find line 751-760 in the revised manuscript. Please clarify.
  • I think the authors should info some more background info regarding different type of vaccines [for example,  IJBS 17(6): 1461-1468].

Author Response

Response to Reviewer 1 Comments

Point 1: According to the authors' response, I won't able to find line 751-760 in the revised manuscript. Please clarify.

Response 1: We apologized for the error of our response. We change occured in the statistical segment. In the manuscript “with track change”, this occure in line 728-738. In the manuscript “without track change” this occure in line 130-140.    

Point 2: I think the authors should info some more background info regarding different type of vaccines [for example,  IJBS 17(6): 1461-1468].

Response 2: Thank you for your suggestion. We provide some more background of the BNT162b2 vaccine, vaccine that was authorized for emergency used for children and adolescent in USA. We add 2 new references, include the journal that you recommended. (In the manuscript “with track change”, this occure in line 269-272. In the manuscript “without track change” this occure in line 52-55.)

Reviewer 2 Report

Thank you,

Author Response

Response to Reviewer 2 Comments

Point 1: Thank you.

 Response 1: Thank you the reviewer for the time and effort you have dedicated to providing insightful comments.

Reviewer 3 Report

  1. In Figure 2, it is difficult to read “small letters”. Authors should make the size letters bigger.
  2. In Figure 3, it is difficult to read “small letters”. Authors should make the size letters bigger.
  3. In Table 2, authors should make corrections from 0.0% to 0%.
  4. To compare the effects of BBIBP-CorV vaccine, authors should use Pfizer-BioNTech COVID-19 Vaccine (also known as COMIRNATY) as control.
  5. Authors should describe in detail about BBIBP-CorV vaccine.

Author Response

Point 1: In Figure 2, it is difficult to read “small letters”. Authors should make the size letters bigger.

Response 1: Thank you for your suggestion. We provided the new Figure with the bigger letters as you recommend.

Point 2: In Figure 3, it is difficult to read “small letters”. Authors should make the size letters bigger.

Response 2: Thank you for your suggestion. We provided the new Figure with the bigger letters as you recommend.

Point 3: In Table 2, authors should make corrections from 0.0% to 0%.

Response 3: Thank you for your suggestion. We corrected the number 0.0% to 0%, as you recommend.

Point 4: To compare the effects of BBIBP-CorV vaccine, authors should use Pfizer-BioNTech COVID-19 Vaccine (also known as COMIRNATY) as control.

Response 4: Thank you for your suggestion. However, we did not have the cohort of Pfizer-BioNTech COVID-19 vaccine for use as control. The BBIBP-CorV vaccine is one of the COVID-19 vaccines WHO lists which was demonstrated safety and efficacy. In addition, the mRNA vaccine platform is the most robust COVID-19 vaccine available now. The immunogenicity of the mRNA vaccine is superior to inactivated vaccine platform. However, this did not mean that the inactivated COVID-19 vaccine did not effective. Moreover, the safety profile of the inactivated vaccine seems to be more than the mRNA vaccine platform, also reported in our findings.     

Point 5: Authors should describe in detail about BBIBP-CorV vaccine.

Response 5: Thank you for your suggestion. We would provide more detail of the BBIBP-CorV vaccine as you suggested (line 254-256 in track change version, 60-62 in clear version).      

Reviewer 4 Report

I am satisfied with changes provided

Round 3

Reviewer 3 Report

This study does not seem to have control. To compare the effects of BBIBP-CorV vaccine, authors should use Pfizer-BioNTech COVID-19 Vaccine (also known as COMIRNATY) as control.

Authors should describe in detail about BBIBP-CorV vaccine. We do not know this vaccine. Which did company make?

Round 4

Reviewer 3 Report

Authors should describe this vaccine in more detail, Authors should add COI with this company.
